

# Airborne Lidar and machine Learning Reveal Decreased Snow Depth in Burned Forests

Arielle Koshkin[1], Adrienne M. Marshall[1]

[1]Hydrological Science and Engineering, Colorado School of Mines, Golden, CO, 80401, USA

*Correspondence to*: Arielle Koshkin (akoshkin@mines.edu)

**Abstract.** Wildfires are increasingly burning higher in elevations well into the seasonal snow zone, altering snow accumulation and melt dynamics. However, limited spatially distributed observations throughout the full snow season have constrained our understanding of how these changes vary across space and time. Here, we assess post-fire snow depth changes across nine basins in California's Sierra Nevada using a machine learning (ML) algorithm, Extreme Gradient

Boosting (XGBoost), trained on 50-m resolution airborne lidar. We develop and apply a novel inferential framework, assuming that the ML algorithm trained on each flight captures the effects of fire on snow depth at the time of acquisition. The median cross-validated (5-fold) RMSE across all acquisitions was 0.23 m. During the accumulation season, the trained ML model predicts smaller post-fire snow depth changes than during the ablation season. Across all 115 acquisitions, 77 % of accumulation-season acquisitions and 98 % of ablation-season acquisitions had a lower basin-wide average predicted

snow depth in burned areas compared to unburned areas. Lower elevations (<2,500 m) consistently exhibited smaller, near-zero post-fire snow depth changes compared to higher elevations (>3,250 m). South- and east-facing slopes experienced the largest negative post-fire snow depth changes. These results illustrate a new inferential approach to assessing fire impacts on snow using lidar-derived snow depth and provide insights to snowpack dynamics in burned forests that are novel in their spatial extent and resolution, as well as ability to discern fire impacts throughout the snow season.



## 1 Introduction

Wildfires are increasingly burning in snow-dominated regions and can alter the timing and magnitude of snowmelt
in ways that vary spatially (Giovando & Niemann, 2022; Koshkin et al., 2025; Smoot & Gleason, 2021). Wildfires in the
seasonal snow zone raise concerns for water management because of the potential for long-lasting impacts of large fires on
mountainous snowpacks (Gleason et al., 2019). While recent work has evaluated fire effects on snow disappearance timing
across the western U.S. (Koshkin et al., 2025), we currently lack an understanding of how these effects vary throughout the
water year across location, elevation, and aspect.

In the western US, the majority of water originates from winter snowpacks (Li et al., 2017). Snowmelt runoff provides
water for downstream users in the spring and early summer when societal, ecological, and atmospheric demands are the greatest
(Bales et al., 2006; Immerzeel et al., 2020; Mankin et al., 2015; Mote et al., 2018). With climate warming, snow-water storage
is declining across the West, threatening water availability (Marshall, Abatzoglou, et al., 2019; Mote et al., 2018). Mid-century
warming is expected to cause spring runoff to be less reliable, earlier, and more episodic, disrupting the predictability of
reservoir inflows and water management (Hale et al., 2022; Livneh & Badger, 2020). With increasing pressure on snow-
derived water resources, understanding seasonal snowmelt timing and magnitude changes are essential for effective water
management in the western US.

At the same time, wildfires have increased in size, duration, and severity, and advanced upslope well into the seasonal
snow zone (Alizadeh et al., 2021; Hatchett, Koshkin et al., 2023; Koshkin et al., 2022; Williams et al., 2022) Koshkin (Alizadeh
et al., 2021; Hatchett, Koshkin et al., 2023; Koshkin et al., 2022; Williams et al., 2022). In the western US, 13 % of the land
area has burned since 1984 (Kampf et al., 2022), which represents a 1150 % increase in area burned from 1984 to 2020
(Williams et al., 2022). In California, there was 10 times more satellite-observed wildfire activity in the seasonal snow zone
in 2020 and 2021 compared to 2000-2019 (Hatchett, Koshkin et al., 2023). It is therefore critical to understand how snow
accumulation and melt dynamics change in burned conditions to inform adaptive watershed management.

Post-fire snow accumulation and melt are influenced by countervailing impacts on energy balance and interception
dynamics (Varhola et al., 2010). Fire can alter net shortwave radiation – often a major contributor to the snow energy balance
(Musselman et al., 2017) – in two ways. First, charred woody debris and black carbon are shed from burned trees onto the
snow surface, decreasing the albedo and increasing the energy absorbed by the snowpack (Gleason et al., 2013, 2019). Second,
more shortwave radiation reaches the snow surface due to reduced canopy shading (Dickerson-Lange et al., 2021). These dual
impacts increase energy inputs to the snow, which increases melt (Kampf et al., 2022; Molotch et al., 2004; Wiscombe &
Warren, 1980). This mechanism of post-fire energy balance is thought to be most prominent in warm, maritime snowpacks
already ripe for melting or during the spring when temperatures and solar radiation increase (Koshkin et al., 2025; Smoot &
Gleason, 2021). However, post-fire canopy loss can also increase snow depth. In an unburned forest, canopy interception can
reduce snow accumulation by up to 40 % relative to non-forested areas because the snow intercepted by the canopy sublimates
into the atmosphere (Harpold et al., 2014; Lundquist et al., 2013, 2021; Roth & Nolin, 2017), but the importance of this effect





can vary widely ((Harpold et al., 2014; Lundquist et al., 2013, 2021; Roth & Nolin, 2017). Similarly, canopy removal in burned forests can reduce interception, potentially increasing post-fire snow depth (Burles & Boon, 2011; Harpold et al., 2014). At high elevations, snow disappearance date (SDD) is delayed post-fire (Koshkin et al., 2025) with deeper peak snow depth and snow water equivalent (SWE) (Giovando & Niemann, 2022; Maxwell et al., 2019), suggesting that the interception effect may
dominate. However, in climatically warmer regions, SDD is earlier (Koshkin et al., 2025), suggesting that increased net shortwave may be more important.

Longwave radiation and turbulent fluxes also affect the post-fire snow energy balance. In unburned forests, longwave radiation inputs to the snow surface come from both trees and the atmosphere. After a fire, the loss of tree cover reduces longwave radiation from trees, which may partially offset the increase in net shortwave radiation, especially under clear-sky
conditions (Burles & Boon, 2011; Seyednasrollah et al., 2013). Post-fire environments also increase snowpack exposure to wind, increasing turbulent heat fluxes (Boon, 2009; Gelfan et al., 2004). Forests help protect snow from wind redistribution, retaining more snow within the forest. Wind can scour snow from open areas without tree cover, reducing post-fire snow depths or increasing their variability (Broxton et al., 2015; Dickerson-Lange et al., 2021; Lundquist et al., 2013; Marshall et al., 2019). These changes in energy balance affect seasonal snow accumulation and melt patterns, complicating the prediction
and modeling of snowmelt timing and magnitude for water resources, especially after a fire.

Most previous research focuses on a single metric to quantify post-fire snowpack effects each winter, such as peak SWE, the date on which peak SWE occurs, or SDD (Giovando & Niemann, 2022; Koshkin et al., 2022; Smoot & Gleason, 2021). When temporal observations are available, they are located at one site or at SNOTEL stations (Broxton et al., 2015; Gleason et al., 2013; McGrath et al., 2023), which are at the point scale only and do not fully represent the surrounding terrain
(Herbert et al., 2024). Therefore, there is a need for spatially and temporally distributed data to capture seasonal variability and identify conditions in which snow depth is increased, decreased, or unchanged in burned conditions. Understanding the inter- and intra-annual variability in snow depth is crucial for modeling post-fire snowpacks, predicting snowmelt timing, and assessing the vulnerability of different basins to loss of snowpack post-fire.

To date, machine learning algorithms and airborne lidar have been underutilized to assess wildfire impacts on snow.
In the Sierra Nevada, California, airborne lidar acquisitions of snow depth are increasingly available at multiple points throughout the snow accumulation and melt season, providing relatively accurate observations of snow depth at 50-m resolution across major watersheds (Painter et al., 2016). These data are increasingly used in the evaluation of snow models and data products (Behrangi et al., 2018; Broxton et al., 2019; Yang et al., 2023). Empirically evaluating fire effects on snow in these watersheds using traditional methods typically requires pre- and post-fire data, and existing methods are challenged
by the irregular timing of acquisitions throughout the snow season. Here, we apply a new statistical framework to avoid both of these problems: we train a machine learning (ML) model on each acquisition to reproduce spatial patterns of snow depth at the time of the acquisition based on underlying topography, land cover, and burn status. We then use the trained model to predict spatially distributed snow depth for each acquisition under hypothetical burned and unburned conditions, differencing these two predicted maps to discern what the model "learned" about the impact of fire on snow for that acquisition. This





method is an example of "explainable artificial intelligence" (XAI), specifically Individual Conditional Expectation (ICE), in which the values of one input feature are varied to assess the impacts on model predictions (Dwivedi et al., 2023). Although prior ICE applications have focused on modifying continuous model features and visualizing the resulting ensemble of predictions (Goldstein et al., 2015), here we instead modify a categorical feature (burn) and assess its effect on model predictions across space. Our approach requires the ML algorithm to accurately reflect the impacts of fire on snow depth within

each acquisition but does not rely on pre- and post-burn data, annual summary statistics, climate forcings, or temporally continuous snow data.

In this study, we use the percent difference in ML-predicted snow depth values for burned and unburned conditions to understand how post-wildfire snowpack changes vary: (1) across the accumulation to ablation seasons, (2) spatially across the Sierra Nevada range, and (3) with elevation and aspect. The results will provide new insight into the spatially and

temporally varying influences of fire on snow accumulation and melt dynamics, leveraging airborne lidar data in a new statistical framework to provide an unprecedented combination of spatial resolution and extent in estimates of post-fire changes in snowpack.

## 2 Methods

This study includes nine snow-dominated basins in the Sierra Nevada, California (Figure 1). These basins were selected based on available lidar data from the Airborne Snow Observatory (ASO) and the occurrence of at least one wildfire between 2015 and 2024. Each basin had at least one year in which lidar was acquired between 2020 and 2024. This five-year study period captured a wide range of hydroclimate conditions, ranging from snow drought (2021) (McEvoy & Hatchett, Koshkin, 2023) to deluge (2023) (Marshall et al., 2024).






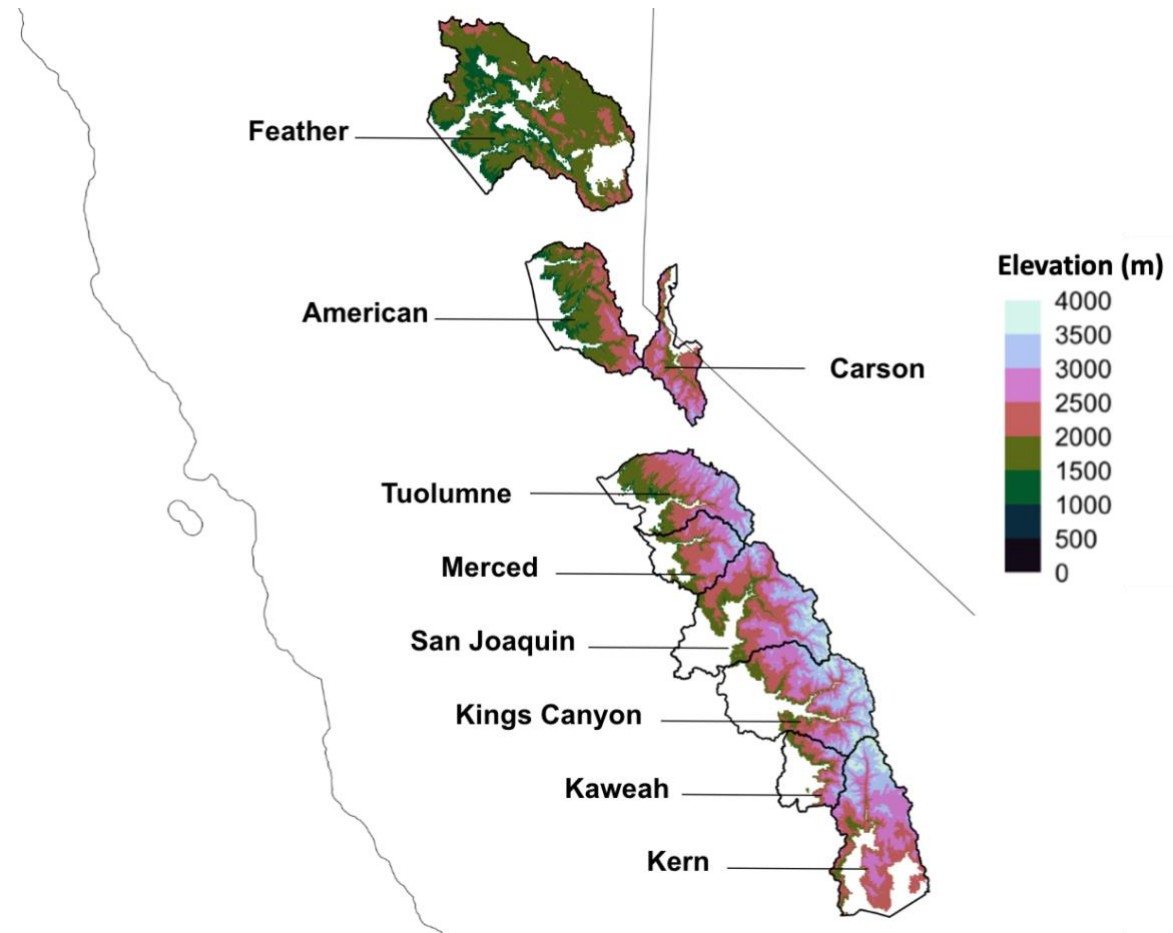

**Figure 1. Map of elevation across the Sierra Nevada**. The basins shown are the nine basins used in this study based on the criteria stated above.

## 2.2 Data

### 2.2.1 Snow Depth

We obtained spatially distributed airborne lidar-derived snow depths from ASO. ASO uses a lidar differencing approach to calculate snow depth by subtracting "snow-off" and subsequent "snow-on" co-located digital elevation models extracted from lidar point clouds. The method produces snow depth rasters at 3 m resolution, which are then aggregated to 50 m resolution (Painter et al., 2016). ASO is flown when water managers procure flights; the temporal resolutions and extents are therefore irregular, with flight frequency ranging from one to six flights per season in each watershed. Here, we use a total of 115 ASO flights over the five-year study period from January to May, totaling more than 2e17 km² (Table S1). The reported mean absolute error (MAE) of the 3-m ASO snow depth product relative to in situ observations is less than 8 cm (Painter et al.,



2016). To account for higher uncertainties in steep terrain due to uncertainties in the viewing angle (Bui & Glennie, 2023), we
removed pixels with greater than 30° slopes. This resulted in a loss of 14.5 % of pixels.

### 2.2.2 Fire Data

Burned pixels were identified using the Monitoring Trends in Burn Severity (MTBS) dataset**,** an interagency dataset produced
by the United States Geological Survey (USGS), United States Forest Service (USFS), and United States Department of
Agriculture (USDA) to construct a long-term dataset of burn severity and extent of large fires across the U.S. from 1984-2024.
Fire perimeters are derived from NASA Landsat data at 30-m resolution by calculating the difference in the Normalized Burn
Ratio (dNBR) using a normalized band ratio of TM 4 and 7 (Eidenshink et al., 2007). We used data from 2015 (5 years before
the first flight) through 2024 for a 10-year span of burn data. Previous studies have shown snowpacks recover from post-fire
impacts within 10 years following a fire (Gersh et al., 2022; Koshkin et al., 2025). We considered a pixel burned if it was
labeled as low, moderate or high burn severity and resampled the rasters to 50 m resolution using the nearest neighbor method.
We used burn severity of low, moderate and high as an input into the machine learning algorithm and 0 if unburned.

### 2.2.3 Forest Cover

We used the dynamic Rangeland Condition Monitoring Assessment and Projection (RCMAP) tree cover time series raster
produced by the Multi-Resolution Land Characteristic Consortium (MRLC) to capture year-to-year changes in canopy cover
(Rigge et al., 2023). This is a 30-m pixel time series (1985-2023) of percent tree cover (0-100 %) for the Western U.S. The
nearest neighbor method was used to resample the rasters to 50-m resolution to match the ASO data. Pixels were filtered to
those with greater than 10 % tree cover to constrain the analysis to forested areas.

### 2.2.4 Peak SWE

Basin-wide peak SWE was calculated to determine whether each ASO flight was acquired during the snow accumulation
(before peak SWE) or ablation season (after peak SWE). For the purpose of calculating peak SWE date, we used daily SWE
data from Snow Data Assimilation System (SNODAS), a 1-km daily product derived from modeling and data assimilation
produced by the NOAA National Operational Hydrological Remote Sensing Center (NOHRSC) (National Operational
Hydrologic Remote Sensing Center, 2004). SNODAS provides a physically consistent framework that integrates airborne,
satellite, and ground station observations to estimate snow variables, including SWE. To calculate basin-wide peak SWE, we
summed daily SWE within each basin and found the date when basin SWE was at its maximum. For each lidar acquisition,
we then calculated the difference between the acquisition date and the date of peak SWE, binning these differences into 30-
day periods to approximate months before or after the date of peak SWE. One limitation is that peak SWE timing varies within
each basin and year, but the simplification to a basin-wide peak SWE date facilitates analysis of results for each acquisition.

### 2.3 Machine Learning



We used a machine learning approach called Extreme Gradient Boosting (XGBoost) to predict snow depth across nine basins in California under hypothetical burned and unburned conditions. This method was selected to develop a
framework in which we could isolate the effect of burn on snowpack dynamics while holding other variables constant. XGBoost was trained separately on each ASO acquisition, using UTM x and y, elevation, aspect, slope, burn severity based on MTBS from 2015-2024, year since burn, and percent tree cover to predict snow depth at 50 m resolution (Figure S1). Areas that burned prior to 2015 were treated as unburned. Crucially, our approach requires XGBoost to capture the spatial patterns of snow depth within each flight, but not to simulate snow accumulation and melt based on climate inputs, nor to have pre-
and post-burn data.

XGboost is an ensemble supervised machine learning approach that applies a gradient-boosting decision tree. The model iteratively builds decision trees and learns from the previous iteration to minimize error and converge towards an optimal set of predictive trees (Chen & Guestrin, 2016). Each tree is learned to minimize the regularized loss and improve the model's predictions while preventing overfitting (Chen & Guestrin, 2016). This approach has become common in remote sensing
(Arabameri et al., 2021; Karthikeyan & Mishra, 2021; Kavzoglu & Teke, 2022) and, more recently, has been successfully applied to develop a daily SWE product (Sun et al., 2024). XGBoost performs well with large datasets that exhibit complex non-linear relationships, is highly robust to outliers, and is less computationally intensive than other machine learning algorithms (Chen & Guestrin, 2016). This algorithm was applied using the "mlr3verse" and "XGboost" packages in R (Chen et al., 2024; Lang et al., 2025).

We applied random search optimization to each model to calibrate the hyperparameters in XGBoost. Random Search optimization is based on decision theory with similar principles to the grid search, but is more time-efficient and easily parallelizable (Kavzoglu & Teke, 2022; Yang & Shami, 2020). The random search selects optimal hyperparameter values to train on within a given configured space until the predetermined threshold value is exhausted (Kavzoglu & Teke, 2022; Yang & Shami, 2020)). This method was selected because of the high optimization performance compared to four other optimization
approaches (Kavzoglu & Teke, 2022).

This optimization approach was applied to all 115 flights individually before running the XGBoost model using the "mlr3randomsearch" in the "mlr3verse" packages in R (Lang et al., 2025). Using a 5-fold cross-validation out-of-sample comparison, the parameters were selected for the model with the lowest cross-validated root mean squared error (CV RMSE). CV RMSE was calculated by comparing the predicted pixels under null conditions to the training data. The hyperparameter
optimization was applied to the learning rate, maximum depth of the decision tree, subsampling rate, a fraction of features to be evaluated at each split, and the number of iterations (to increase training speed and reduce overfitting) (Bentéjac et al., 2021).

**2.4 Estimating Snow Depth Differences in Burned vs Unburned Forests**

We used the fitted models to predict snow depths for hypothetical burned and unburned conditions for all pixels,
where the year since burn was set to 1, burn severity was set to 4 (high burn), and canopy cover was set to 10 % for burned





pixels. We evaluated the difference between predicted hypothetical burned and unburned values by calculating the percent difference in predicted snow depth:

$$\Delta\text{SD} = \frac{pSD_\text{B} - pSD_\text{UB}}{pSD_\text{UB}} * 100 ,\tag{1}$$

where $\Delta\text{SD}$ is the percent difference in snow depth, $pSD_\text{UB}$ is the predicted snow depth if the pixel was unburned, and $pSD_\text{B}$ is
the predicted snow depth if the pixel was burned. This approach assumes that the model learned the post-fire snow depth variability accurately, which we evaluate by comparing RMSE in burned and unburned conditions (Figure 2). Variations in $\Delta\text{SD}$ were qualitatively compared based on the elevation, aspect, and month since peak SWE on both basin-wide and 50-m spatial scales. Basin-wide averages ($\text{SD}_\text{BW}$) were calculated by summing burned and unburned snow depth values separately across all pixels for each acquisition within the seasonal snow zone and calculating the basin-wide percent difference; this
method was selected rather than averaging pixelwise percent differences to avoid the influence of pixels with small denominators that could result in anomalously large percentage values. The seasonal snow zone was defined by pixels that had more than 30 consecutive days of SWE in each of the 5 acquisition years.

## 3 Results

### 3.1 Model validation results

The median cross-validated (5-fold) CV-RMSE across all flights was 0.23 m (IQR: 0.15 m). Burned pixels had a lower CV-RMSE with a smaller IQR compared to unburned pixels (Figure 2). Specifically, the median CV-RMSE for burned pixels was 0.11 m (IQR: 0.08 m), whereas unburned pixels have a median CV-RMSE of 0.23 m (IQR: 0.14 m). These results may be influenced by the disproportionate ratio of burned to unburned pixels, as unburned pixels account for 97 % of the
dataset, which increases overall variability. This was consistent across water years as well: 2023 accounted for 38 % of pixels and had the largest magnitude and variability in CV-RMSE (median = 0.31 m; IQR: 0.31 m) (Figure 2a). Model performance for burned pixels declined during the accumulation season and gradually improved during the ablation season. In contrast, CV-RMSE for unburned pixels was more consistent across months since peak SWE (Figure 2b). The basin size did not appear to strongly influence model performance. For example, the Feather basin, which has the largest area, showed the best fit for non-
burned pixels, while the Kaweah basin, the smallest in the dataset, had moderate model performance (Figure 2c).



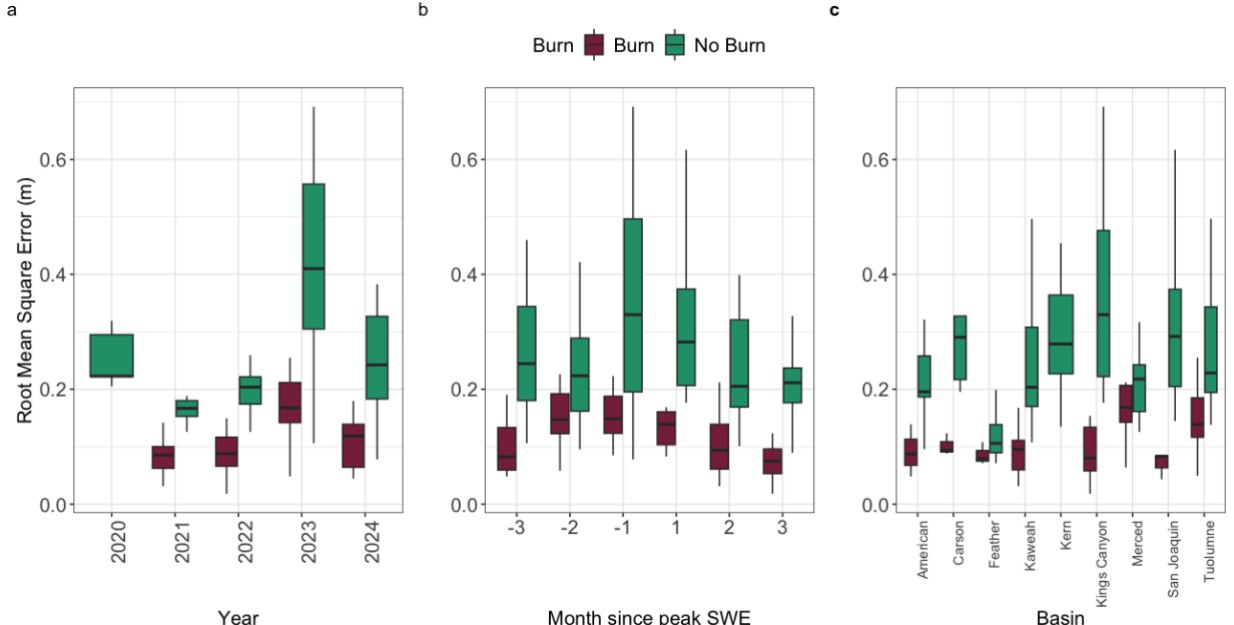

**Figure 2.** Cross validated root mean squared error (CV-RMSE) for burned (red) and unburned (blue) pixels for every acquisition by (a) year of flight, (b) month since peak SWE, and (c) basin.

## 3.2 Post-fire snow depth changes increase during accumulation and decrease during melt

Machine learning simulations of snow depth for predicted burned and unburned conditions indicated that the basin wide percent difference in post-fire snow depth ($\Delta SD_{BW}$) was predominantly negative in both the accumulation season (before peak SWE) and ablation season (after peak SWE), but the magnitude of change was much larger in the ablation season (Figure 3; Figure S2). Across all 115 acquisitions, 77 % of accumulation acquisitions had a lower basin-wide average predicted snow depth in burned than unburned conditions, compared to 98 % during melt. The median $\Delta SD_{BW}$ was -7.8 % (IQR: 11.3) across all seasons but was -1.75 % (IQR: 3.24) in the accumulation season, compared to -10.0 % (IQR: 10.4) during the melt season. The accumulation seasons had a smaller $\Delta SD_{BW}$ and less variability compared to the ablation season. About half (48 %) of melt-season acquisitions had $\Delta SD_{BW}$ values lower than -10 %, while only 11 % of accumulation season $\Delta SD_{BW}$ values exceeded ±10 % (Figure 3; Figure S2).

The contrast between accumulation and ablation season effects is further emphasized in individual months. The largest $\Delta SD_{BW}$ occurred two and three months after peak SWE, with median values across all basins and years of -11.5 % (IQR:9.7) and -12.9 % (IQR:10.3), respectively. In contrast, the difference was minimal two months (-1.25 %; IQR:2.7) and one month before peak SWE (-1.75 %; IQR: 4.9). During accumulation, $\Delta SD_{BW}$ decreased from three months prior to two months prior to peak SWE, while in the ablation season, $\Delta SD_{BW}$ declined each month following peak SWE. Variability was




substantially higher in the ablation season and increased throughout the snow season (Figure 3). Basin-wide average interannual temperature and snowfall variability did not appear to be a large driver of this heterogeneity (Figure S3).

 Within individual basins, $\Delta SD_{BW}$ patterns are more variable, suggesting local topographic or climatic factors influence post-fire snowpack response (Figure 3b; S2). The Kings Canyon and Kaweah basins exhibited the strongest seasonal contrast in $\Delta SD_{BW}$, with smaller losses during accumulation and the largest losses during melt (median $\Delta SD_{BW}$ was 13-17 %

across ablation months in the two basins). In contrast, the Kern showed little median seasonal change ( ± 1 % difference). During accumulation, the Carson was the only basin with positive $\Delta SD_{BW}$ with a median of +2.2 %. In all basins except the Kern and Tuolumne, $\Delta SD_{BW}$ exceeded -10 % during the ablation season.

 By three months post-peak SWE, the Kern showed a 11 % increase in basin-wide snow depth under burned conditions, while the Feather showed a 23 % decline. These patterns highlight the large spatial variability in post-fire snowpack

response, which is not clearly explained by latitude or timing alone (Figures 3, 4, S2, S4).

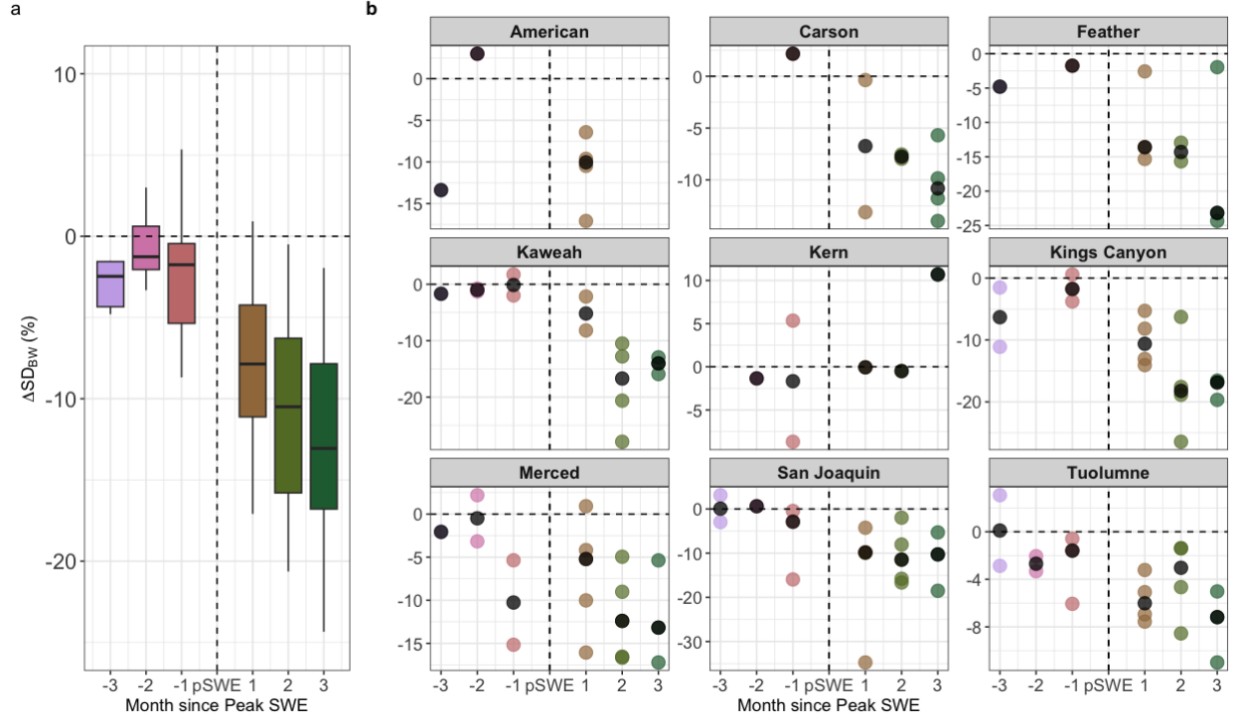

**Figure 3.** (a) Basin-wide average percent difference in snow depth ($\Delta SD_{BW}$) by month relative to peak SWE. (b) $\Delta SD_{BW}$ by basin and month relative to peak SWE. Each point represents one acquisition. Black dots are the median $\Delta SD_{BW}$ for each month since peak SWE for each

basin. Negative values on the x-axis indicate acquisitions before peak SWE; positive values indicate acquisitions after. Months are binned in 30-day intervals.

## 3.2 Post-fire snow depth varies spatially



Post-fire changes in $\Delta SD_{BW}$ show substantial spatial heterogeneity across basins, years, and months since peak SWE,
particularly at the 50-m scale (Figure 4; Figure S4). Across all acquisitions, the median $\Delta SD_{BW}$ for the northern basins
(American, Carson, Feather) was nearly double that of the southern basins (Tuolumne, Merced, Kings Canyon, San Joaquin,
Kern, Kaweah), with $\Delta SD_{BW}$ values of -9.8 % and -5.7 %, respectively. However, when broken down by season, the median
$\Delta SD_{BW}$ becomes more closely aligned, especially during the accumulation season, with $\Delta SD_{BW}$ values of -1.7 % for the
northern basins and -1.8 % for the southern basins. The ablation season, however, exhibits a larger discrepancy between the
two regions: mean $\Delta SD_{BW}$ in the northern basins is -11 %, compared to -9.9 % in the southern basins.

At the 50-m scale, pixel-wise median percent difference in post-fire snow depth ($\Delta SD_P$) variability is heterogeneous
across each acquisition for each basin. Flights in the ablation season had smaller spatial variability compared to those in the
accumulation season. Out of 115 flights, 13 had more than 80 % of $\Delta SD_P$ with the same sign, with all but two of these flights
occurring in the ablation season. Not all pixels exhibited the same patterns within a given basin or season (Figure 4; Figure
4S). For example, many acquisitions had evidence of more positive $\Delta SD_P$ estimated along rivers, while others had clear
elevation gradients within the acquisition.

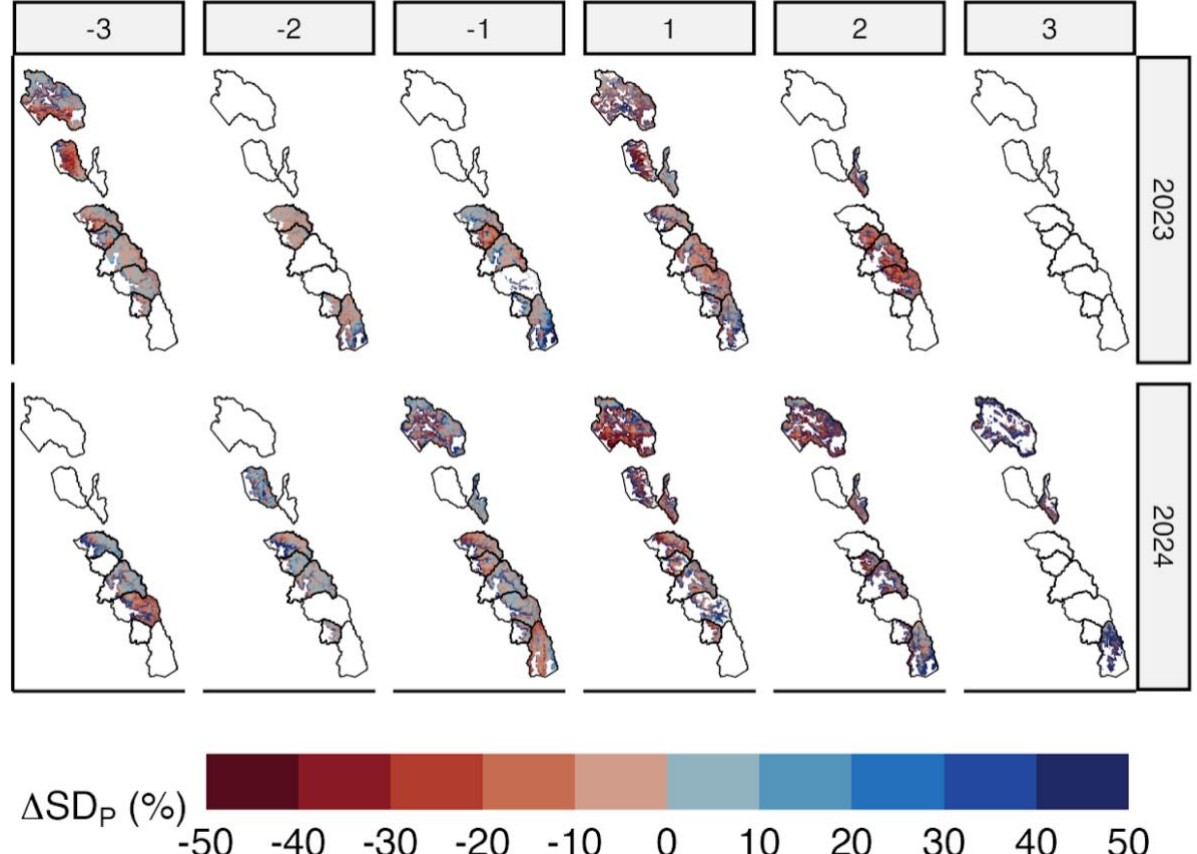





**Figure 4.** Spatial maps of post-fire snow depth percent difference (ΔSD$_P$) for water years 2023 and 2024—the years with the most lidar acquisitions—across nine study basins by month since peak SWE. See Figure S4 for maps from other years.


### 3.3 Elevation and aspect influence heterogeneous post-fire snow depth changes

Despite large variability within elevation bands, lower elevations (< 2000 m) consistently exhibited smaller and near-zero ΔSD$_P$ compared to higher elevation (>3000 m), where ΔSD$_P$ was more strongly negative (Figure 5). Low elevations had greater variability in ΔSD$_P$, and the spread in ΔSD$_P$ narrowed as elevation increased. Seasonal patterns further highlight
elevation-driven trends, with larger variability during the ablation season compared to the accumulation season, especially at lower elevation (Figure 5b). During the accumulation season, especially 1-2 months before peak SWE, ΔSD$_P$ at low elevations was often near-zero or positive, whereas moderate to high elevations were consistently negative during both the accumulation and ablation seasons. At the lowest elevation (<2000 m), 37 % of ΔSD$_P$ during the accumulation season were negative compared to 48 % during ablation. At high elevation, in both seasons, over 85 % of ΔSD$_P$ were negative.

The magnitude of negative ΔSD$_P$ increased with elevation, particularly during the ablation season, with the largest negative ΔSD$_P$ occurring above 3000 m. However, the largest seasonal shift in median ΔSD$_P$ occurred at mid elevation (2000-3000 m), where median ΔSD$_P$ changed from -0.6 % in accumulation to -7.0 % in ablation. Low-elevation (<2000 m) had the smallest seasonal change, from -+2.5 % in accumulation to +2.0 % in ablation. While mid elevations exhibited large seasonal shifts in ΔSD$_P$, often transitioning from near positive to negative ΔSD$_P$ around peak SWE, high elevations remained
consistently negative across both seasons. However, one month after peak SWE, mid-to-high elevations had smaller ΔSD$_P$ compared to lower elevations and high elevations had more consistent ΔSD$_P$ throughout ablation. (Figure 3b). The proportion of the basin area at high or low elevation was not correlated with ΔSD$_{BW}$. Across all acquisitions, the Feather and Kings Canyon basins had the largest ΔSD$_{BW}$; however, they have contrasting elevation profiles – the Feather is the lowest elevation basin with no area above 3000 m, while Kings Canyon is among the highest with 36 % of the area above 3000–suggesting that
basin-wide differences are not strictly elevation-driven (Figure S5).





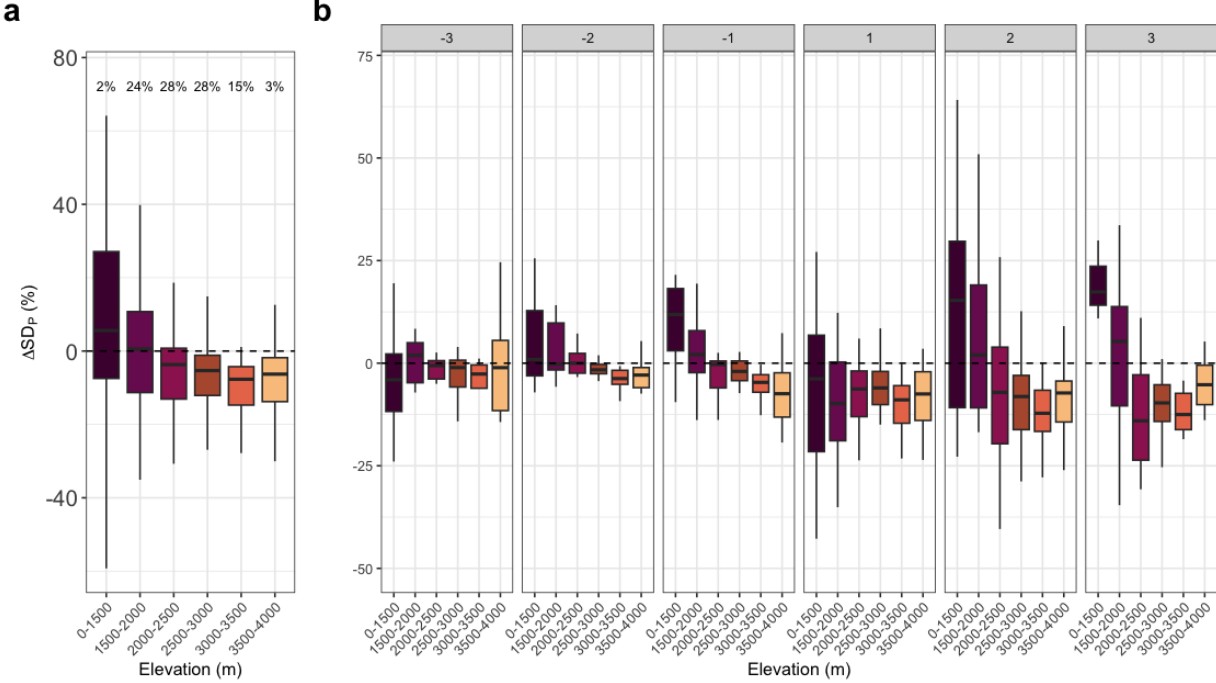

**Figure 5.** (a) Interquartile range of post-fire snow depth percent difference ($\Delta SD_P$) by elevation band across all basins; text denotes fraction of basin area within each elevation band. (b) Same as in (a) split out by month since peak SWE.

Aspect also influences $\Delta SD_P$ heterogeneity, particularly at high elevation and during the ablation season (Figure 6).
During the accumulation season, $\Delta SD_P$ magnitudes were smaller and more consistent across aspects. Northwest-facing slopes
exhibited some positive $\Delta SD_P$ prior to peak SWE, but the median $\Delta SD_P$ across all aspects remained negative, ranging from -
5.8 % (north) to -8.6 % (south).

However, during the ablation season, aspect had a strong influence on $\Delta SD_P$. South-, west-, and east-facing slopes
had more negative $\Delta SD_P$, with median $\Delta SD_P$ exceeding -12 %, whereas northern slopes had a median of -8.6 % (Figure 6a).
The largest median $\Delta SD_P$, occurred 2–3 months post-peak SWE on south- and east-facing slopes, exceeded -10 %, while north-
facing slopes had a median of -6.0 %. The smallest median $\Delta SD_P$ was observed two months prior to peak SWE on north- and
east-facing slopes (+0.35 % and -0.72 %, respectively), with western and southern slopes ranging from -2 % to -1 %. Overall,
south- and east-facing slopes drove the largest snow reductions during ablation, whereas north- and west-facing slopes were
less affected.

Aspect and elevation also interacted to affect $\Delta SD_P$. South-and east-facing slopes exhibited the largest negative $\Delta SD_P$,
particularly at low elevations (< 1500m) and high elevations (>3500 m), with median $\Delta SD_P$ ranging from -9.3 % to -17.5 %.
West- and north-facing slopes had a smaller median $\Delta SD_P$ at both low and high elevations, ranging from -3.0 % to -8.4 %. At
mid-elevations (2000–3000 m), the largest $\Delta SD_P$ was concentrated on south slopes (-7.2 %), whereas northern slopes at mid-




elevation showed the smallest median $\Delta SD_P$ (-3.9 %) (Figure 6b). Aspect influence varies across elevation bands, and large negative $\Delta SD_P$ values were most heavily concentrated on east-facing slopes.

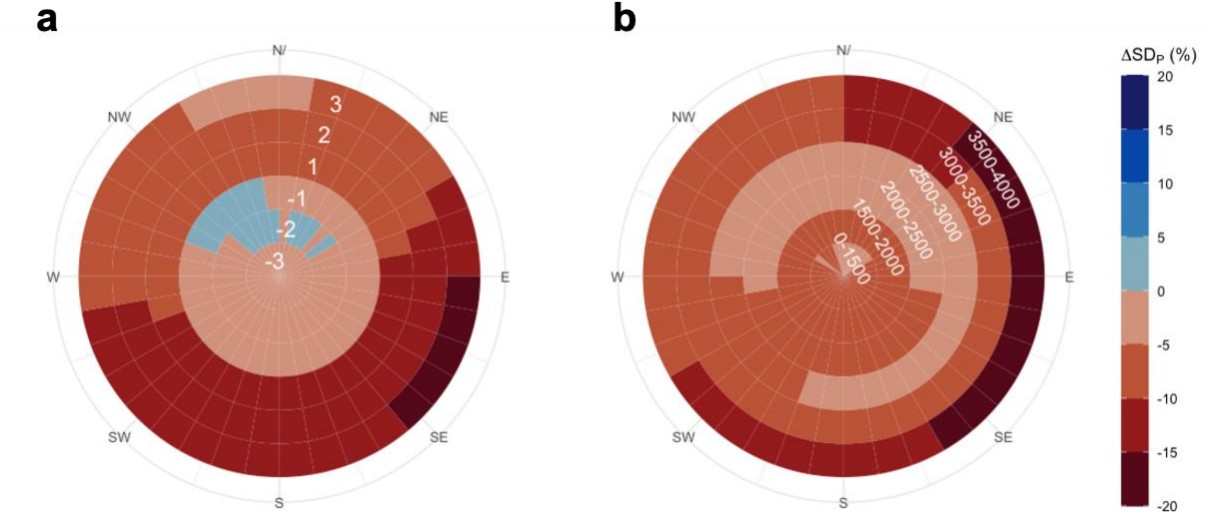

**Figure 6.** Median post-fire snow depth percent difference ($\Delta SD_P$) by aspect. (a) $\Delta SD_P$ by aspect for each month relative to peak SWE. (b)
As in (a), shown by elevation band (m).

## 4 Discussion

### 4.1 Fire effects on snow depth vary throughout the snow season and across elevation and aspect

ML-predicted snow depth is lower in burned compared to unburned forests throughout the snow season, with much
larger impacts in the ablation than accumulation season. The variability in these differences varies spatially across elevation bands, aspects, and basins, with the largest effects at high elevations and southern aspects. This pattern suggests an increased rate of snowmelt under burned conditions after peak SWE and reflects a seasonally dependent shift in dominant snowpack energy balance processes following wildfire. Low elevation during the ablation season had the largest variability in post-fire snow loss.

The variability in post-fire snowpack changes is likely driven by a trade-off between increased accumulation from reduced canopy interception and enhanced melt from lower albedo and greater shortwave radiation. During accumulation, reduced canopy interception in burned forests mitigates snow loss, especially when snowpacks are cold and storms are frequent (Hatchett, Koshkin et al., 2023). These conditions help retain snow depth in burned forests and mitigate substantial losses. Our analyses suggest smaller losses in snow depth during the accumulation season. Conversely, during spring melt, when solar
radiation is higher and albedo resets occur less frequently, burned snowpacks absorb more energy due to black carbon deposition from charred trees, leading to accelerated melt (Gleason et al., 2013). These observations support the idea that small shifts in energy balance in snowpacks with minimal cold content can rapidly initiate melt (Jennings et al., 2018; Lundquist et





al., 2013), consistent with previous findings of earlier snow disappearance and predominantly lower, earlier peak SWE post-fire (Giovando & Niemann, 2022; Kampf et al., 2022; Koshkin et al., 2025; McGrath et al., 2023; Smoot & Gleason, 2021).

This energy balance trade-off could explain why there is less snowpack loss in burned forests during accumulation compared to the ablation season.

This seasonal contrast is further explained by elevation. The largest post-fire snow loss occurred at higher elevations, particularly 2-3 months post-peak SWE, well into spring months. These areas also retain snow the longest, extending ablation into a period of high solar radiation when melt is accelerated (Musselman et al., 2017). Reduced albedo in a burned forest,

combined with increased incoming solar radiation, may cause these snowpacks to absorb more energy, accelerating melt and, in turn, snow loss compared to unburned forests. The effect is especially pronounced on south- and east-facing slopes, which receive greater solar exposure. In contrast, lower-elevation areas melt earlier in the season, when solar radiation is less intense, so the differences between burned and unburned forests are smaller; this potential mechanism is supported by the observation that elevation differences are relatively minimal earliest in the accumulation season and is consistent with similar findings

evaluating the impact of dust-on-snow (Réveillet et al., 2022). However, these findings at high elevation are inconsistent with previous work that found that high elevation burned forests had smaller changes in snow disappearance timing compared to low elevation (Koshkin et al., 2025). This contrast could be an artifact of the definitions of high elevation as Koshkin et al. (2025) found larger advance in post-fire snowmelt timing in the Sierra Nevada compared with the rest of the intermountain west, especially in northern basins, which is consistent with our findings. This finding may also be subject to uncertainty

among different datasets and statistical methods and should be examined further in future work.

Aspect further exacerbates differences in accumulation and melt season dynamics following fire – a finding that is novel to this study, given the advantages of high-resolution, spatially distributed data used here. North-facing slopes, which generally have higher snow retention due to low radiation loads (Blanken & Barry, 2016), had the smallest amount of snow loss in burned forests, with slightly deeper snow than in unburned forests in the accumulation season. Snow accumulation

from canopy loss in these areas may insulate the snowpack from substantial albedo-driven energy degradation when cold contents are negative (Dickerson-Lange et al., 2021; Harpold et al., 2014; Lundquist et al., 2013), leading to smaller snowpack losses. Conversely, south and east-facing slopes experienced the largest snow depth reductions in burned forests, consistent with higher radiation loads and more rapid energy gain from albedo degradation (Blanken & Barry, 2016; Wiscombe & Warren, 1980). This is especially true at high elevation 2-3 months after peak SWE (well into spring). This is congruent with

previous findings of high-elevation south-facing burned slopes in the Colorado Rocky Mountains reaching peak SWE 22 days earlier than north-facing slopes (Reis et al., 2024). The results imply that post-fire changes on north- and west-facing slopes may be mitigated by canopy change, particularly early in the season, while south- and east-facing aspects may be more dominated by albedo change exacerbated by high radiation loads.




### 4.2 Methodological innovations

Our results provide a quantitative analysis of seasonal, spatial and topographic variability in post-fire snow, integrating a process-based inference framework with a predictive machine learning algorithm using explainable AI methods

that are novel in snow hydrology. The XGBoost algorithm used has become common in remote sensing gap-filling applications (Arabameri et al., 2021; Karthikeyan & Mishra, 2021; Kavzoglu & Teke, 2022) and has recently been applied to create a daily reanalysis SWE product (Sun et al., 2024). Similarly, process-based inference has previously been used in snow hydrology, for example, to examine snowfall intensity impacts on snow storage (Marshall et al., 2020) and to quantify changes in post-fire snow disappearance date across the western US (Koshkin et al., 2025). Some other hydrology studies have combined

machine learning and process-based inference using models to understand flooding characteristics (Schmidt et al., 2020) and flash-flood susceptibility (Abedi et al., 2022). Using a neural network and random forest, Schmidt et al. (2020) showed that inferential ML had higher predictive accuracy than a linear regression and reflected physical hydrological principles accurately, which generally aligns with our findings. However, they also identified differences in the results from two ML algorithms (analogous to equifinality issues in process-based models), suggesting a potential need for future work examining

the robustness of our more detailed results to the choice of ML algorithm.

Our approach merged ML and inferential statistics, allowing for new understanding of the relative influence of topography and seasonality on post-fire snow changes, while not being limited by the need for pre-and post-fire snow data. While classical statistical approaches inspect regression slopes to obtain effect sizes (e.g. Luce et al., 2014), we effectively infer the effect size by comparing model-predicted values in burned and unburned conditions. Our framework demonstrates

that predictive machine learning can complement process-based reasoning, particularly in cases where observational data are sparse, baseline records are missing, or the processes of interest are highly nonlinear. On average, our models had a median RMSE of 0.23 m (ranging from 0.07m to 0.7 m) of snow depth. Previous work has assessed error of model products based on SWE, rather than snow depth. A rough conversion of our snow depth RMSE to SWE based on modeled density derived from ASO SWE data suggests a median SWE RMSE of 0.12 m across all 115 flights. This is comparable to model errors

from SNODAS, REC-Parbal and National Water Model SWE (Yang et al., 2023). Our RMSE values were similar to or lower than previous ML-based efforts to simulate snow depth (Cartwright et al., 2022; Daudt et al., 2023; Herbert et al., 2025), though each of these studies have slightly different aims and training data provided to the models.

Compared to traditional statistical approaches, machine learning offers several advantages for process inference. Classical statistical models, such as linear regression or generalized additive models, require restrictive assumptions about

data structures and form of relationships and can struggle with nonlinear responses and complex interactions among variables (Wood, 2017). In contrast, XGBoost and other machine learning algorithms can flexibly model nonlinearities and higher-order interactions without pre-specifying functional forms, is robust to outliers, and can handle complex datasets. These strengths are particularly valuable in studies assessing wildfire impacts on snow, where the effects of burn severity, canopy loss, and terrain features on snow depth are spatially heterogeneous and interact in non-linear ways. By combining

high-resolution ML predictions with process-based analysis, we bridge a methodological gap in snow hydrology, allowing





for robust inference about the relative influence of topography, seasonality, and fire on snowpack loss in areas where pre-fire observational data may be unavailable or not comparable to post-fire data.

### 4.3 Hydrologic consequences of fire impacts on snow accumulation and melt

405       Although not evaluated in this study, the fraction of the basin that burns could influence the hydrologic significance of post-fire snow depth changes, especially at the basin scale. In this study, basin-wide averages are calculated by treating the entire basin as hypothetically burned. The San Joaquin basin, where the Creek Fire burned over 40 % of the basin in 2020 (*CAL FIRE*) also had some of the largest post-fire snow depth changes in this study. The magnitude of these changes is an important consideration for water managers when a burn occurs in the sub-alpine portion of their basin, where impacts are

disproportionately concentrated. However, downstream hydrologic effects may be partially mitigated by unburned or low-elevation areas within the watershed, which can provide a spatial counterbalance to the most impacted areas.

      Understanding topographic and seasonal controls on snow loss in burned forests is crucial for assessing impacts on water availability, runoff timing, and flood risk. Following a wildfire, peak streamflow often increases (Williams et al., 2022) and can occur 1–50 days earlier (Pirani & Coulibaly, 2025). High-elevation snowpacks act as natural water storage and

contribute disproportionately to the snowmelt hydrograph, accounting for up to 70% of streamflow generation during the melt period (Sprenger et al., 2024). In burned forests, accelerated snow loss at high elevations decreases snow storage and shifts melt contributions earlier in the season. Earlier peak SWE (Smoot & Gleason, 2021) and snow disappearance date (Koshkin et al., 2025) in burned forests, combined with reduced snow depth, pushes melt earlier in the season when we expect snowpacks to yield less-concentrated streamflow pulses (Bazlen, 2025). However, in burned forests the lower albedo could drive faster

melt, potentially producing a concentrated spring melt pulse resulting in increased post-fire peak flow. This elevation-driven acceleration of post-fire snowmelt alters both the timing and magnitude of runoff, creating challenges for water managers reliant on predictable high-elevation snowmelt to sustain water supply through summer. This challenge may be further exacerbated by the fact that the largest observed increase in burned area is above 2500m (Alizadeh et al., 2021), where snow may be particularly difficult to monitor (Serreze et al., 1999).

425       The effects of fire can persist for up to 6–10 years following a fire before albedo and energy balance processes stabilize (Gersh et al., 2022; Koshkin et al., 2025). In the short term, burned forests may have limited snow loss during the accumulation season, but melt faster, driven by the post-fire albedo changes exacerbated in the spring. This is congruent with our findings from the ML-predicted snow depth one year after a fire. However, in the long term, as albedo effects wane, the legacy of canopy loss may lead to sustained increases in snow accumulation post-fire, particularly in deeper snowpack in high-

elevation areas (Harpold et al., 2014). Future work could expand our methods to evaluate trajectories of post-fire snow impacts across multiple years.

### 5 Conclusion





Wildfires disrupt patterns of snow accumulation and melt, challenging our ability to predict water availability in
burned watersheds. The direction of change is predominantly negative, but the magnitude varies by season, aspect, elevation
and basin characteristics. A decrease in snow depth post-fire, especially after peak SWE, could lead to earlier runoff timing,
especially in high-elevation basins. Since basin topography is a compilation of different elevations and aspects, the
consequences of burned forests for snowpack dynamics on a basin-wide scale could be impacted by where the fire occurs in
the basin. Increased snow depths in one part of a burn scar could counterbalance sharp declines in another, potentially evening
out basin-wide effects. Wildfire impacts on snowpack dynamics are complex and highly variable, highlighting the need to
continue to leverage different methods (machine learning, statistical and process-based models, in situ observations) and
datasets (field campaigns, airborne lidar, satellites) to evaluate the impacts of wildfire on snowpacks on relevant spatial (plot,
basin-wide, mountain range) and temporal (daily, monthly, yearly) scales.

**Code availability:** All code used to produce the machine learning models and analysis is available at
https://zenodo.org/records/14728778?preview=1&token=eyJhbGciOiJIUzUxMiJ9.eyJpZCI6IjExOTUxZTFmLTAxZTAtND
U1Yy1iZjQ5LWFjZDZjYTc0YWFmMSIsImRhdGEiOnt9LCJyYW5kb20iOiIyNTU3MzZmZjg5MTNkMTY4ODYzOTAz
NDk2NzY2NjE0YiJ9.dWwlABJChb-
DhjbjILQ7JgmO_sRXj5EeXS0WH2KUl9LWwLlE_ott8Z63l_gRH1SdguPDurUez2SAaiK2m7lF2. We used R programming
language version 4.3.1. for all data processing, analysis, and graphical representation. We will make all code publicly
accessible upon publication.

**Data availability:** All data used in this project are publicly available and accessible. The data used in this work came from
SNODAS, https://nsidc.org/data/g02158/versions/1, the monitoring trends in burn severity (MTBS) https://www.mtbs.gov/,
Airborne Snow Observatory, Inc. https://data.airbornesnowobservatories.com/.

**Author Contributions:** AK: conceptualization, data curation, formal analysis, methodology, code development, validation,
visualization, writing – original draft, writing – review and editing. AM: conceptualization, funding acquisition, methodology,
writing – review and editing, supervision


**Acknowledgments:** This work used Jetstream2 GPU at Indiana University through allocation EES250101 from
the **Advanced Cyberinfrastructure Coordination Ecosystem: Services & Support** (ACCESS) program, which is
supported by U.S. National Science Foundation grants #2138259, #2138286, #2138307, #2137603, and #2138296 (Boerner
et al., 2023). Comments from Jessica Lundquist and an anonymous reviewer improved an earlier version of this manuscript.


**Financial Support:** AK and AM were funded by Dr. Marshall's start-up funding at the Colorado School of Mines.



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
