# Peer review of "Airborne Lidar and machine Learning Reveal Decreased Snow Depth in Burned Forests"

_EGUsphere, 2025_

## Referee Comment (RC1)

**Review of "Airborne Lidar and Machine Learning Reveal Decreased Snow Depth in Burned Forests" for *The Cryosphere**

E. N. Boardman, eli.boardman@mountainhydrology.com (non-anonymous, feel free to contact with any questions)

**Summary**

The authors train machine learning (ML) models to reproduce lidar-based snow depth measurements across a large spatial domain, several years, and different points in the season. Using the trained models, the authors predict the effect of forest fires on snow depth by perturbing the model predictor variables to represent counterfactual burned and unburned conditions. The authors analyze spatial and temporal variability in these model predictions and provide a process-based interpretation of their data-driven findings.

**Strengths**

The fundamental premise of the study is fairly obvious and simple, yet elegant and interesting, which is perhaps the best kind of study. The application of "big data" (>100 lidar surveys) to post-fire snow hydrology is important and novel. This is the kind of paper I would definitely cite in the future, and it provides a clear launching pad for similar future investigations (perhaps comparing additional models, finer resolutions, or other geographic areas). The efforts towards a physical interpretation of ML results is also commendable. The numeric results are interesting both from a basic science standpoint and are directly transferrable to water/forest management questions and future model validation applications.

**Main Comments**

**(1) Treatment of non-forest alpine areas**

It is unclear to me how the authors are currently treating the large portions of the study watersheds that are above treeline. In the Sierra Nevada, there are many thousands of km2 covered by talus and granite bedrock. In these alpine areas, it would be meaningless to talk about the effect of burning a non-existent forest.

It seems like the authors may have dealt with this issue by masking out pixels with <10% forest cover, but persistent usage of terminology like "basin-wide" makes this unclear. Additionally, several of the figures clearly show predicted  $\Delta SD$  (snow depth difference in burned forests) across entire basins, even in high alpine regions without forests, which is confusing and physically implausible.

I see two possibilities: (A) the current study is masking out the non-forest pixels already, in which case this should be clarified throughout (and the maps should be similarly masked) to avoid confusion, or (B) the current study is predicting  $\Delta SD$  everywhere, even in barren alpine regions, which should be corrected.

A related concern is the possibility of pixels that are initially forested and then go to 0% forest cover after a major fire (quite common with the RCMAP dataset used here). How are these pixels handled, and how are newly deforested pixels discriminated from never-forested pixels?

**(2) Informal use of "inference" language**

Throughout the manuscript, the authors refer to their methods as "inference," but the study does not seem to contain any formal inferential framework. I understand that in the machine learning world, "inference" is used equivalently to "prediction." However, in my view this is an unfortunate artifact of the informal ML lingo that should not be perpetuated in the natural sciences.

It seems like the so-called "process-based inference" (Discussion section) is informally derived from the authors' expert knowledge and literature review rather than quantitative inference. To call something "inference," I would want to see a quantitative framework defining prior and posterior distributions, a likelihood function, etc. Maybe a compromise would be to call it "process-informed reasoning" or something? Things like Bayesian ML do exist, and should be distinguished from the informal inferential procedure used here (not that informal inference doesn't have a strong legacy in hydrology...cf. Beven's GLUE).

Specific suggestions for rewording are included in my detailed comments. If the authors want to persist in using "inference," I think this should be very carefully and explicitly caveated where it appears (Abstract, Methods, Discussion, etc.) to acknowledge that the type of "inference" performed here does not yield statistical confidence intervals, posterior distributions, hypothesis tests, etc. Alternatively, the study could be reworked to leverage the large wealth of hybrid Bayesian-ML approaches, which could enable true inference in a statistical sense, but this would probably require quite a bit of additional work, so it's probably easier to change the language.

**(3) Spatial autocorrelation and cross-validation**

The out-of-sample predictive accuracy of the trained model is obviously of paramount importance for this study, since that is how the snow depth difference is calculated. However, the current approach to model validation is potentially impacted by spatial autocorrelation, and a more robust approach to train/test data partitioning would greatly enhance believability.

Section 2.3 refers to "cross validation" and "out-of-sample comparison," but it is unclear how the separate train/test sets are derived for these comparisons. Lacking any specific explanation, I assume that all pixels within a single ASO survey were randomly sampled for training the trees comprising each XGBoost model. However, with 50 m grid cells covering a complete spatial area, most grid cells are adjacent to many other grid cells with near-identical predictors (nearly the same elevation/aspect/slope/forest/fire history). Thus, I wonder whether the model is actually learning meaningful information, or whether it is just interpolating between pixels. For example, if the pixels at (x, y-1) and (x, y+1) are in the training dataset, it is quite easy to predict the pixel at (x, y) in the test dataset through simple spatial interpolation. Thus, I am concerned that the cross-validation error metrics could be confounded by spatial autocorrelation within the gridded snow depth data.

The way I have handled similar problems myself is by separating train/test datasets using a large grid, e.g., alternating 1 km blocks of training and test pixels. In the authors' application, the model is asked to predict the effect of hypothetical fires at locations that are many kilometers away from any historical fire location. Thus, the authors should demonstrate that the model is capable of predicting snow depth at a similar distance from the training data, not just the next pixel over. For each ASO survey, I suggest imposing a 1 km (or larger) grid of train/test regions, training the model only within some of these 1 km grid regions, and testing the model predictive accuracy on the other out-of-sample 1 km grid regions. This would provide more of a true out-of-sample estimate of predictive accuracy since snow depth autocorrelation is much lower at kilometer scales compared to 50 m. This would also overcome some of my concerns about using UTM x-y as a predictor variable (namely, that the model can just memorize the snow depth map by interpolating between known training coordinates).

**Detailed Comments**

Abstract: "trained on 50-m resolution airborne lidar [snow depth data?]"

Abstract: The sentence beginning "During the accumulation season" seems out of place to me—I would expect some broader statement first, like "On average, snow depth is X% lower in burned areas."

Abstract: "basin-wide average predicted snow depth in burned areas" is unclear to me—presumably it's implausible for the entire basin to burn, since much of it is just alpine rock? Maybe just take out "basin-wide" in the abstract until this can be clarified later.

Abstract: Sometimes the lower elevations actually have an increase in burned snow depth if I understand correctly? Might be worth adding that to the "smaller, near-zero changes."

Lines 26-27: "long-lasting impacts of large fires on mountainous snowpacks" Maybe add "mountainous snowpacks and snow-dominated water resources" to broaden the implications? We just had a study accepted at *HESS* might be relevant, which shows that the Creek Fire increased annual San Joaquin runoff by as much as 18% during a drought year, with substantial implications for water management in that basin:

Boardman, E. N., Boisramé, G. F. S., Wigmosta, M. S., Shriver, R. K., and Harpold, A. A.: Improving Model Calibrations in a Changing World: Controlling for Nonstationarity After Mega Disturbance Reduces Hydrological Uncertainty, EGUsphere [preprint], <a href="https://doi.org/10.5194/egusphere-2025-1877">https://doi.org/10.5194/egusphere-2025-1877</a>, 2025

Line 39: This string of citations seems to have some typographical errors and repeats

Line 81: "relatively accurate" might be an understatement given the nominal snow depth uncertainty of < 1 cm when aggregated to 50 m resolution (cf. ASO survey reports). Also, the observations are at 3 m resolution, which seem to be typically distributed along with the 50 m data in the survey zip folders.

Line 85: Not just the irregular timing of surveys—the interannual weather variability also massively confounds pre/post-fire analyses, which is why counterfactual modeling experiments (as done here with ML, or using process-based models) are the norm for disturbance attribution. Interannual variability in albedo caused by different levels of atmospheric deposition could also be salient for a snow study.

Line 98: "changes vary" is confusing wording to me—it's technically correct, and I know what is meant, but maybe consider rewording for clarity?

Line 107: "five-year study period" is confusing because of the two preceding date ranges—it seems like there are two periods under consideration (2015-2024 and 2020-2024), and it's not immediately clear how these two periods are being used differently.

Figure 1: I think this would be really cool as a multi-panel figure, with a second panel showing the most recent year burned (colors filled within each fire perimeter) and perhaps additional panels showing RCMAP forest cover and ASO snow depth (perhaps the maximum pixel-wise snow depth across all acquisitions?)

Line 121: Some years/basins have even more than 6 flights (2016 in the Tuolumne comes to mind), so perhaps just say "from one to six or more flights per season"

Line 122: The total area of "2e17 km2" seems to be a typo, because this is physically implausible. For reference, the land area of Earth is 1.5e8 km2, which is 9 orders of magnitude

smaller. Given 115 ASO flights, with a maximum basin area of say 10,000 km2 (i.e., the Feather), the total surveyed area cannot be more than 1e6 km2. Similar problem in Table 1: why are the units in  $10^{15}$  km2? This is again physically implausible. I would recommend just listing the basin areas in km2, since the area of each basin falls in the range of ~1,000 to ~10,000 km2.

Line 138 (RCMAP section): I would add a sentence outlining the basic methodology for how Rigge et al. derive these data. Also, be careful of using the year immediately before a fire. I haven noticed that major fires often show up as "ghosts" in the prior year's dataset (i.e., the September 2020 Creek Fire perimeter is visible in the 2020 RCMAP data as a slight reduction in canopy cover, even though there was no fire effect during the 2020 snow season). They might have fixed this in a later data release—not sure—but definitely worth visually checking some of the immediately pre-fire years. Additionally, I have noticed that the canopy cover in RCMAP is often reduced to 0% or 1% after a major fire—how is this being handled in the >10% masking? Specifically, do pixels that are initially forested and change to 0% tree cover after a fire still get included in the ML training? Are the authors using a static mask across all years (in which case, what years are used to define the 10% threshold?) or are the authors using a dynamic mask for each separate year (in which case, how do newly treeless pixels get handled?)

Line 145 (Peak SWE section): might be worth checking pairs of ASO flights from before/after SNODAS peak SWE to validate that all of the post-peak-date ASO surveys have less basin-total SWE than the pre-peak-date surveys.

Section 2.3: A word search doesn't return any results for "counterfactual," which I think is a key word related to the approach here. Specifically, it would enhance the clarity in my mind if the authors specified that the ML model is used to predict snow depth in counterfactual burned/unburned scenarios, which eliminates the issue of interannual variability for the burneffect attribution. Might also help with future keyword search optimization.

Line 161: Using UTM x and y as predictors is potentially problematic. With a large enough model, it could just memorize the snow depth for each unique x-y location. Currently, there seems to be no justification for why these x-y predictors are used, or what the plausible physical interpretation would be. I suspect this is being used to capture synoptic scale weather patterns, i.e., "the north side of the basin gets more snow," but in that case, why not use climatological maps from PRISM or similar? At minimum, I would like to see some explanation of what these x-y predictors are intended to capture, and a more robust spatial sensitivity test (see major comment on spatial autocorrelation and validation). Ideally, I would like to see if similar results could be reproduced using something other than x-y, such as interpolated climatological maps, or even a smoothing kernel applied to the ASO snow maps (to capture preferential deposition or other weather effects that might be missing from climatological maps).

Line 163: is it realistic to treat areas that burned prior to 2015 as unburned, given the (slow?) growth rate of alpine conifer forests? I realize that the albedo effect is probably small after that much time, but I'm not convinced that the interception recovers completely that fast.

Line 170: I think the "success" of a given SWE product is subjective and depends on the intended use; I would just take out that word and say "applied to develop a daily SWE product."

XGboost section overall: I would like to see some discussion of convolutional neural networks or other SOTA neural approaches to spatial ML like GANs or VAEs. In particular, convolutional nets can use the local spatial context for predictions (i.e., drifts downwind of terrain features, forest edges, gaps, etc.). This is probably less important in forest regions, which tend to have more uniform snowpacks—perhaps this could be stated as a justification for using a simpler tree-based approach, combined with the computational efficiency (though backpropagation is also pretty efficient). I'm also curious why the XGBoost prediction features don't include any topographic metrics beyond elevation/aspect/slope—what about topographic roughness, position index, upwind angle, etc.?

Line 184: "under null conditions to the training data" what does this mean? Even as someone who fancies myself a bit of a ML researcher at times, I've never heard this phrase, and I suspect it will be foreign to many non-ML snow scientists too. Please elaborate.

Line 189: I think this section should be substantially expanded since it gets at the real crux of the whole study—the comparison of counterfactual burned/unburned. Specifically, a few things are unclear to me currently: if this comparison was done "for all pixels," does this include pixels that have never been forested (above treeline)? If it's only for the masked pixels with >10% forest cover, see comment on RCMAP pixels that decrease to near 0% after fire. Also, why only set burn severity to high? It seems like for minimal additional effort, the authors could add an additional interesting comparison between the effects of high/medium/low severity and number of years post-fire.

Line 199: "basin-wide" is a bit misleading I think, assuming that the comparison is only made within the forested region? Maybe area-average would be a more precise term, or "basin average within the forested region." Otherwise, I think this carries the implication that a 10% change in post-fire forest snow equates to a 10% change in basin-total snow, which is not true (potentially much of the snow is above treeline in some basins).

Line 204: See major comment on spatial cross-validation.

Figure 2: A lighter shade of green might make these boxes easier to distinguish in black-and-white. Also, something is weird with the legend—"Burn" seems to be repeated both sides of the red box.

Line 220: the increase/decrease terminology is confusing, since it implies a directionality in  $\Delta$ SD, whereas I think the intended meaning is just the magnitude of this effect.

Figure 3: This is great! My only though is that "Peak" might be clearer than "pSWE" for the horizontal axes labels.

Figures 4 and S4: I don't understand why the  $\Delta SD$  maps extend all the way to the highest reaches of the Tuolumne, Merced, San Joaquin, Kings, etc., which is an extreme alpine area devoid of forest. Are the  $\Delta SD$  values calculated everywhere, or just within the forested region? It wouldn't make sense to talk about the  $\Delta SD$  of these high alpine slopes. If the main  $\Delta SD$  stats in the paper are masked just to the forested region (is this what the 10% canopy cover threshold is for?), this should also be reflected in Figure S4 to avoid confusion.

Line 273: How many burned forest pixels exist above 3000 m? (Or 3500 m for that matter—Figure 5). Is this a sufficient sample size to justify these comparisons? In the Illilouette, we seem to have an upper fire line around 2600 m. Either way, it would be helpful to know the distribution of burn area training data with elevation.

Figure 6: It looks like the 0-1500 m elevation range has a positive median  $\Delta$ SD per Figure 5a, but I don't see any positive  $\Delta$ SD for the 0-1500 m range in Figure 6b. Am I misunderstanding something?

Line 320: "variability in these differences varies" awkward wording, how about "these differences vary spatially"

Line 367 / Section 4.2: I'm not sure I would go so far as to call this process-based, when the process implications seem to just be assumed from prior literature. Something more process-based might be calibrating a model like SnowPALM to ASO, then running it in counterfactual burned/unburned scenarios.

Line 381: I don't see any inferential statistics. Where is this inference performed, and what are the associated hypothesis tests, likelihood functions, credible intervals, etc.? Not all modeling experiments count as "inference" in my opinion.

Line 385: I think the choice of "process-based reasoning" (used here) is much more accurate than "process-based inference" (used elsewhere).

Lines 386-392: This comparison of RMSE values is unfair. The other SWE datasets discussed are not directly trained on the target data. In theory, given a large enough model and enough predictors, the approach used here ("predicting" SWE within individual flights) should achieve RMSE ~0, since the true answer is used as the training data.

Line 393: The authors seem to pose a false dichotomy between "traditional statistical approaches" and "machine learning," when in fact there is a substantial overlap. Relegating "traditional statistics" to just mean "linear regression" ignores a huge body of prior work on advanced nonlinear statistical inference. For instance, Gaussian Process regression is a fully Bayesian ML method that does not require pre-specifying functional forms, most Bayesian sampling algorithms use the same automatic differentiation method that is at the core of all neural networks, etc. Moreover, techniques like variational Bayes can be interpreted equally well using either traditional statistics or machine learning conceptualizations (<a href="https://en.wikipedia.org/wiki/Variational\_Bayesian\_methods">https://en.wikipedia.org/wiki/Variational\_Bayesian\_methods</a>). I suggest that the authors substantially reword or remove this section in light of the considerable overlap and intermingling between "traditional statistics" and "machine learning," rather than just dismissing "traditional statistics" as basically antiquated.

Line 404 (Hydrologic impacts section): I would add more references to literature specifically addressing the hydrological impacts of fire in the Sierra Nevada, not just the snow impacts. In addition to our study of the Creek Fire water yield effects cited earlier, here are a few more:

Abolafia-Rosenzweig, R., Gochis, D., Schwarz, A., Painter, T.H., Deems, J., Dugger, A., Casali, M. and He, C. (2024), Quantifying the Impacts of Fire-Related Perturbations in WRF-Hydro Terrestrial Water Budget Simulations in California's Feather River Basin. Hydrological Processes, 38: e15314. <a href="https://doi.org/10.1002/hyp.15314">https://doi.org/10.1002/hyp.15314</a>

Boisramé, G. F. S., Thompson, S. E., Tague, C., & Stephens, S. L. (2019), Restoring a natural fire regime alters the water balance of a Sierra Nevada catchment. *Water Resources Research*, 55, 5751–5769. https://doi.org/10.1029/2018WR024098

Roche JW, Goulden ML, Bales RC. Estimating evapotranspiration change due to forest treatment and fire at the basin scale in the Sierra Nevada, California. *Ecohydrology*. 2018; 11:e1978. https://doi.org/10.1002/eco.1978

Line 407: "entire basin as hypothetically burned" even huge areas of granite talus above treeline?

Conclusion: This is a nice concise summary, well done.